# Predicting mortality among patients with liver cirrhosis in electronic health records with machine learning

**Aixia Guo[1], Nikhilesh R. Mazumder[2,3], Daniela P. Ladner[3,4], Randi E. Foraker[1,5]** *

**1** Institute for Informatics (I2), Washington University School of Medicine, St. Louis, MO, United States of America, **2** Division of Gastroenterology, Northwestern Memorial Hospital, Chicago, IL, United States of America, **3** Northwestern University Transplant Outcomes Research Collaborative (NUTORC), Comprehensive Transplant Center, Feinberg School of Medicine, Northwestern University, Chicago, IL, United States of America, **4** Division of Transplant, Department of Surgery, Northwestern Medicine, Chicago, IL, United States of America, **5** Department of Internal Medicine, Washington University School of Medicine, St. Louis, MO, United States of America

* randi.foraker@wustl.edu

**Data Availability Statement:** As the data set contains potentially identifying and sensitive patient information, we cannot share these data publicly without permission. To request the data, please

## Abstract

### Objective

Liver cirrhosis is a leading cause of death and effects millions of people in the United States. Early mortality prediction among patients with cirrhosis might give healthcare providers more opportunity to effectively treat the condition. We hypothesized that laboratory test results and other related diagnoses would be associated with mortality in this population. Our another assumption was that a deep learning model could outperform the current Model for End Stage Liver disease (MELD) score in predicting mortality.

### Materials and methods

We utilized electronic health record data from 34,575 patients with a diagnosis of cirrhosis from a large medical center to study associations with mortality. Three time-windows of mortality (365 days, 180 days and 90 days) and two cases with different number of variables (all 41 available variables and 4 variables in MELD-NA) were studied. Missing values were imputed using multiple imputation for continuous variables and mode for categorical variables. Deep learning and machine learning algorithms, i.e., deep neural networks (DNN), random forest (RF) and logistic regression (LR) were employed to study the associations between baseline features such as laboratory measurements and diagnoses for each time window by 5-fold cross validation method. Metrics such as area under the receiver operating curve (AUC), overall accuracy, sensitivity, and specificity were used to evaluate models.

### Results

Performance of models comprising all variables outperformed those with 4 MELD-NA variables for all prediction cases and the DNN model outperformed the LR and RF models. For example, the DNN model achieved an AUC of 0.88, 0.86, and 0.85 for 90, 180, and 365-day mortality respectively as compared to the MELD score, which resulted in corresponding

contact the Washington University Human Research Protection Office by mail at MS08089-29-2300, 600 S. Euclid Avenue, Saint Louis, MO 63110 or by phone at 314-747-6800 or by email hrpo@wustl.edu.

**Funding:** Dr. Mazumder was supported by NIH grant T32DK077662. The other authors received no specific funding for this work.

**Competing interests:** The authors have declared that no competing interests exist.

AUCs of 0.81, 0.79, and 0.76 for the same instances. The DNN and LR models had a significantly better f1 score compared to MELD at all time points examined.

## Conclusion

Other variables such as alkaline phosphatase, alanine aminotransferase, and hemoglobin were also top informative features besides the 4 MELD-Na variables. Machine learning and deep learning models outperformed the current standard of risk prediction among patients with cirrhosis. Advanced informatics techniques showed promise for risk prediction in patients with cirrhosis.

## Background and significance

Cirrhosis of the liver is a leading cause of morbidity and mortality in the United states, causing 40,000 deaths each year [1]. The vast majority of patients with cirrhosis have subclinical disease, however once their disease progresses they often rapidly decompensate and are at high risk of morbidity, mortality, and poor quality of life [2, 3]. The current method for predicting mortality in sick patients relies on the Model for End Stage Sodium (MELD-Na) score, a modified logistic regression model developed in 2002 that accurately predicts 90 day mortality at high scores and can help triage treatment and monitoring [4, 5].

Unfortunately, while accurate at high scores and short follow up, the MELD-Na score is not as predictive at lower scores and longer time spans [6–8]. Furthermore, the vast majority of patients with cirrhosis have missing labs to calculate a MELD-Na score or have low MELD-Na scores, with 93% having a MELD-Na of less than 18 [9, 10]. Thus, an alternative method to predict mortality for the cohort of patients with cirrhosis at large is needed.

One possible reason for the conventional MELD-Na score fails to predict patient outcomes may be due to the complex biological relationship among non-linear and multi-dimensional variables present in medicine [11]. Deep learning algorithms have been successfully used in some healthcare applications due to their ability to effectively capture informative features, patterns, and variable interactions from complex data [12–16]. One 2006 study demonstrated that an artificial neural network performed better than MELD in predicting 3-month mortality among 400 patients with end-stage liver disease [11]. Another study in 2018 investigated 12–24 hour mortality prediction among 500 patients critically ill patients with cirrhosis by logistic regression (LR) and long short-term memory (LSTM) neural networks [17]. The limitations of these studies were that the cohorts included relatively few patients and predicted short-term mortality rather than long-term mortality, which allows for interventions that may change the outcomes.

### Objective

The goal of our study was to assess if deep learning and machine learning techniques can provide an advantage in predicting 90 day, 180 day, and 365 mortality in patients with cirrhosis.

## Materials and methods

### Ethics statement

The need for informed consent for this study, which used existing patient records, was waived by the Institutional Review Board (IRB) of Washington University in Saint Louis (IRB ID # 202006212).

## Method

**Data source and study design.**   In this study, we utilized electronic health record (EHR) data from a large academic liver transplant center. Our institution partnered with MDClone [18, 19] (Beer Sheva, Israel) for the data storage and retrieval. MDClone platform is a data engine by storing EHR medical events in a time order for each patient. Queries can be built to pull computationally-derived or original EHR data from the platform. Patients in the EHR were included if they had an initial diagnosis code of liver cirrhosis in the period 1/1/2012 through 12/31/2019 and were 18 years of age or older at first diagnosis. The cohort was identified based on the following International Classification of Disease codes: K76.6 Portal hypertension, K76.1 cardiac/congestive cirrhosis of liver, K74.60 Unspecified cirrhosis of liver, B19.21 Unspecified Viral Hep C with hepatic coma, K70.30 Alcoholic cirrhosis of liver, K74.69 Other cirrhosis of the liver, K70.31 Alcoholic cirrhosis of liver with ascites, I85.00 Esophageal varices, I86.4 gastric varices, K76.7 Hepatorenal syndrome, K65.2 Spontaneous bacterial peritonitis, K74.3 Primary Biliary Cirrhosis, E83.110 pigmentary cirrhosis (of liver), E83.01 Wilson's Disease, I85.01 Esophageal varices with bleeding, K76.81 Hepatopulmonary syndrome similar to previous studies [20, 21]. Time of inclusion was defined as the first occurrence of any of these codes.

**Primary outcome.**   The primary outcome was all-cause mortality ascertained by the medical record. We performed analyses within three specific timeframes 90 days, 180 days, and 365 days.

**Feature extraction.**   Features were measured at or before the time of first diagnosis code. They included patient demographics, laboratory data, and information on last hospitalization. The selected features are predictive for mortality in patients with liver cirrhosis. Baseline demographic characteristics such as age, race and ethnicity, and laboratory features collected from blood such as serum aspartate aminotransferase, alanine aminotransferase, and total bilirubin were all informative predictors for mortality predictions in patients with liver cirrhosis [22, 23]. We included features that had more than 10% non-missing values, otherwise we discarded them. For included features, missing values were imputed using the multiple imputation [24] for the continuous variables and the mode for categorical variables. We also compared the multiple imputation strategy with mean imputation strategy by using mean for continuous variables. The results for mean imputation strategy were shown in Supplementary materials. For features with multiple values, only the closest value prior to inclusion was used. Using this method, 41 features were incorporated into models. The included features and possible value examples for each feature were listed in **S1 Table**.

**Statistical analysis.**   We assessed and compared a deep learning model (DNN [25]) with two machine learning models, random forest (RF) [26] and LR [27], to predict mortality and compared this to using the four variables in the MELD-Na model (serum sodium, International Normalized Ratio (INR), Creatinine, and Total Bilirubin).

For each model, we utilized 5-fold cross validation. Due to the low overall rate of mortality, Synthetic Minority Over-sampling Technique (SMOTE) [28] for Nominal and Continuous was used to deal with imbalanced classes by oversampling positive patients in each fold for training but the original distribution was maintained in the validation fold. We studied 18 total prediction cases in terms of 3 different windows (365 days, 180 days and 90 days), two different groups of features (41 and 4 variables), and three different models (DNN, LR, and RF). We utilized the receiver operating curve, overall accuracy, sensitivity, specificity, precision, and F1-score to evaluate the performance of models for each prediction case.

We then investigated the feature importance to understand the reason of better performances for models with 41 features compared to those of 4 variables. The importance of each

feature was quantified by each of the three trained models. We studied all the three models to confirm that the three methods could lead to consistent results. The coefficients for each input variable retrieved from the LR model was used to measure the feature importance for each input feature. The mean decrease impurity importance of a feature by the trained RF model was used to assess feature importance of RF model. We used the "iNNvestigate" [29] package with gradient to calculate feature importance for the DNN model.

Our DNN was comprised of an input layer, 4 hidden layers (with 128 nodes each) and a scalar output layer. We used the Sigmoid function [30] at the output layer and ReLu function [31] at each hidden layer. Binary cross-entropy was used as loss function and Adam optimizer was used to optimize the models with a mini-batch size of 512 samples. The hyperparameters were determined by optimal accuracy using a grid search method from 2 to 8 for hidden layers of network depth, 128 and 256 for hidden layer dimensions, 64, 128, and 512 for batch size.

We performed a grid search of hyperparameters for RF and LR models by five-fold cross validation. We searched the number of trees in the forest for 200, 500, and 700, and we considered the number of features for the best split according to auto, sqrt, and log2. For the LR model, we searched the norm for L1 and L2 norm in penalization, and the inverse value of regularization strength for 10 different numbers spaced evenly on a log scale of [0, 4].

The RF model was configured as follows: the number of trees in the RF was set 500; the number of maximum features that could be used in each tree was set as the square root of the total number of features; the minimum number of samples at a leaf node of a tree was set as 1. The LR model was configured as follows: the L2 norm was used in the penalization, i.e., the variance of predicted value and real value of training data; the stopping criteria was set as $1.0*10{-4}$; the inverse of regularization strength, which reduces the potential overfitting, was set as 1.0.

**Fig 1** shows our flowchart of our work. Analyses were conducted using the libraries of Sci-kit-learn, Keras, Scipy, and Matplotlib with Python, version 3.6.5 (2019).

## Results

### Cohort characteristics

During the study period, 34,575 patients met the inclusion criteria and their characteristics are listed in **Table 1**. Patients were primarily white, with a mean age of 60 years, with a mean MELD-Na score of 12.3. Approximately 5% (n = 1,784), 6% (n = 2,217), and 8% (n = 2,775) of patients died at 90, 180, and 365 days after inclusion.

### Results of prediction models

**Fig 2** shows the prediction performance for the prediction cases of mortality within 365 days, 180 days, and 90 days by using all the 41 variables and only the 4 variables in MELD-Na model. In all 3 cases, all 3 models consistently indicated that performances with all 41 variables outperformed the cases of using only 4 variables used in the MELD-Na model. Among the 3 machine learning models, DNN model had the best performance (better than LR and RF model) in all the prediction cases with only MELD variables and RF had the best performance in the case of using 41 variables. The average AUC was 0.88 (0.86 and 0.85) for DNN model, 0.80 (0.77 and 0.74) for LR, and 0.90 (0.88 and 0.86) for RF model in the case of 90-day (180-day and 365-day) prediction for the case of 41 variables. In each chart of **Fig 2**, the model name plus MELD-Na in the legend means only the 4 variables in the MELD-Na model were inputs, otherwise, if only the model name is listed, the model used all 41 variables as inputs. For example, DNN Mean AUC in the legend refers to the case of using all 41 variables in DNN model, and DNN MELD-NA Mean AUC refers to the average AUC in the case of DNN model

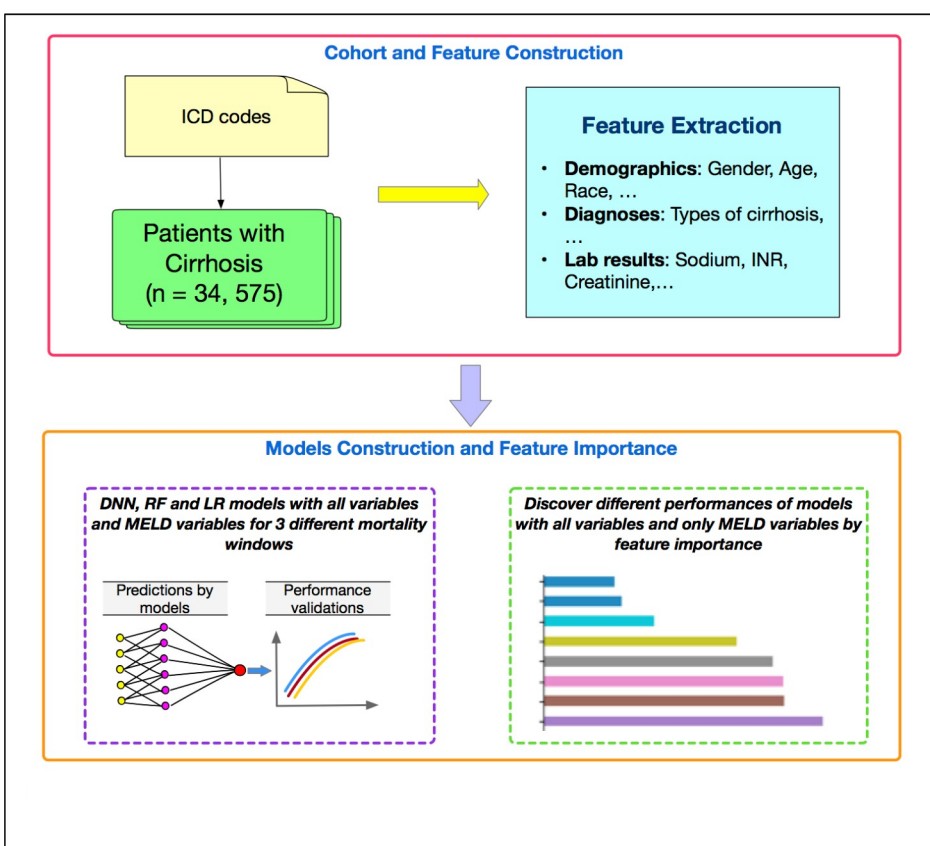

**Fig 1. The flowchart of our work.**

using only the 4 MELD-Na variables in the 5-fold cross-validation. **S1 Fig** shows the prediction performance for the case with mean imputation strategy.

**Table 2** shows other metrics, i.e., overall accuracy, precision, recall, F1-score, and specificity for 3 models by using all variables and 4 MELD-Na variables. In all the cases, DNN and RF models achieved similar higher values compared to the LR model in terms of F1-score. Although RF and DNN achieved similar values of F1-score, DNN model achieved higher recall/sensitivity compared to RF model. All the metrics improved by using all the 41 variables instead of only the 4 MELD-Na variables. Prediction metrics were more accurate for the case of mortality at 90 days, compared to the other two cases of 180 days and one year. **S2 Table** shows the same metrics results with the mean imputation strategy. It is clinically useful to consider different tradeoffs of sensitivity and specificity for the specific clinical application. We have also conducted the analysis considering 10 different tradeoffs, i.e., 0.05, 0.1, 0.2, 0.3, 0.4, 0.6, 0.7, 0.8, 0.9, 0.95. The results of models with all 41 variables were summarized as **S3 Table**.

In order to investigate why the performance decreased when only using the 4 MELD-Na variables compared to using all of the variables, we investigated the feature importance obtained from the three trained models (**Fig 3**). **Fig 3A** was obtained from DNN model, **Fig 3B and 3C** were based on RF and LR models. The three figures consistently showed that the 4 MELD-Na variables were in the top features among all the variables, which indicated that the 4 variables were important and informative. However, besides these 4 variables, other features such as alkaline phosphatase values, Alanine aminotransferase values, hemoglobin values, and hospital admission start date (date difference in days between diagnosis of liver cirrhosis and

**Table 1. Characteristics [mean (SD) or n (%)] of our study populations.**

| Patients | Mean (SD) or n (%) |
|---|---|
| Total patients N | 34,575 |
| Mortality within 365 days n (%) | 2,775 (8.0) |
| Mortality within 180 days n (%) | 2,217 (6.4) |
| Mortality within 90 days n (%) | 1,784 (5.2) |
| Age | 60.5 (14) |
| Gender | |
| Female | 17,600 (50.9) |
| Male | 16,973 (49.1) |
| Race | |
| White | 26,790 (77.5) |
| Black | 5,438 (15.7) |
| Other/unknown | 2,347 (6.8) |
| Ethnicity | |
| Not Hispanic or Latino | 23,156 (67.0) |
| Hispanic or Latino | 313 (0.9) |
| Unknown | 11,106 (32.1) |
| BMI | 29.0 (7.1) |
| INR | 1.3 (0.6) |
| Sodium | 138.2 (3.9) |
| Creatinine | 1.13 (1.06) |
| Total bilirubin | 1.3 (3.2) |
| Hemoglobin | 12.3 (2.3) |
| Potassium | 4.1 (0.5) |
| Bicarbonate | 24.5 (4.6) |
| MELD score | 11.5 (6.2) |
| MELD-Na score | 12.3 (6.5) |

previous hospital admission start dates) were also top features, which meant they might also be informative and play an important role in the predictions.

Fig 3 shows the case of 365-day mortality prediction. The other cases (90-day and 180-day mortality) had the same trends and similar results.

## Discussion

In this study, we utilized 8-year EHR data that were synthesized and deidentified by the MDClone platform to identify patients with liver cirrhosis to predict their mortality within 365, 180, and 90 days from the first diagnoses of liver cirrhosis by machine learning and deep learning models. We also investigated the most important features by ANOVA for understanding the better performance of models with 41 variables than those of 4 variables in the MELD-Na model. We finally investigated the performances of models with 11 different cut-off values to discover changes in model performance.

Our results indicated that the deep learning model DNN can effectively predict the mortality within 90, 180, 365 days of patients compared to LR and RF models in all prediction cases and are superior to the four variables in the MELD-Na score alone. Although the 4 variables used in MELD-Na model were among the top most informative features, other features such as hemoglobin, alkaline phosphatase, alanine aminotransferase, and time since recent hospitalization were also top features and might play an important role in mortality prediction.

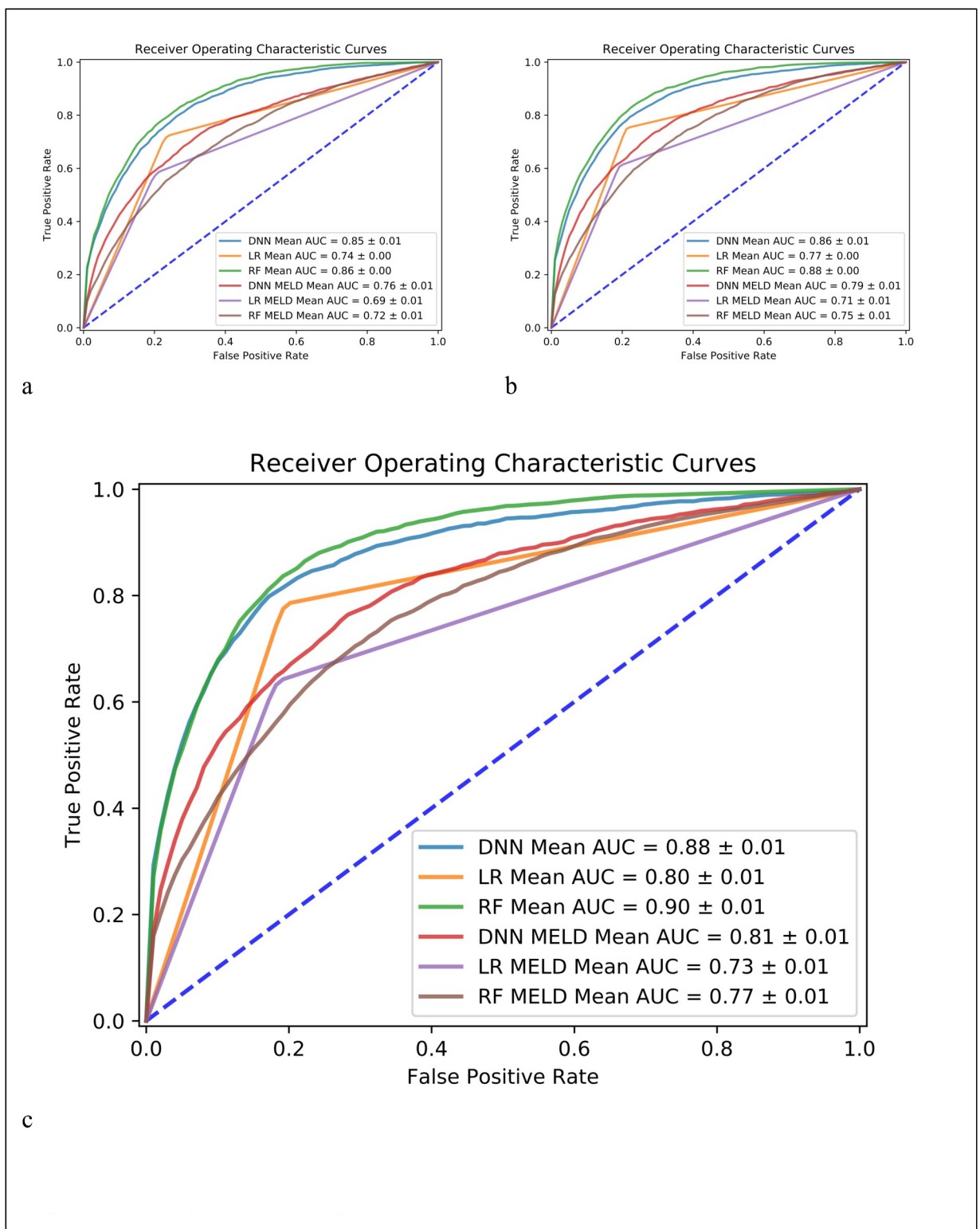

**Fig 2. Prediction performance by deep neural network (DNN), random forest (RF) and logistic regression (LR).** Figure a, b, c is for the case of mortality within 365 days, 180 days, and 90 days, respectively.

**Table 2. Prediction metrics [n (%)] of 3 period cases for 3 machine learning models.**

| Models | Period | Accuracy | Precision | Recall | F1-Score | Specificity |
|---|---|---|---|---|---|---|
| | (days) | Mean(std) | Mean(std) | Mean(std) | Mean(std) | Mean(std) |
| DNN | 365 | 0.83(0.01) | 0.27(0.0) | 0.65(0.04) | 0.38(0.01) | 0.85(0.01) |
| (all variables) | 180 | 0.86(0.02) | 0.26(0.02) | 0.64 (0.03) | 0.37(0.02) | 0.88(0.02) |
| | 90 | 0.90(0.02) | 0.30(0.05) | 0.63(0.04) | 0.40(0.04) | 0.92(0.02) |
| LR | 365 | 0.77(0.01) | 0.21(0.0) | 0.72(0.01) | 0.33(0.01) | 0.77(0.01) |
| (all variables) | 180 | 0.79(0.0) | 0.19(0.0) | 0.75(0.0) | 0.31(0.0) | 0.79(0.0) |
| | 90 | 0.81(0.01) | 0.18(0.0) | 0.78(0.03) | 0.29(0.01) | 0.81 (0.01) |
| RF | 365 | 0.92(0.0) | 0.47(0.04) | 0.37(0.02) | 0.41(0.02) | 0.96 (0.0) |
| (all variables) | 180 | 0.93(0.0) | 0.46(0.03) | 0.40(0.02) | 0.43(0.02) | 0.97 (0.0) |
| | 90 | 0.94 (0.0) | 0.43(0.01) | 0.41(0.02) | 0.42(0.01) | 0.97(0.0) |
| DNN | 365 | 0.78(0.02) | 0.20(0.01) | 0.59(0.04) | 0.30(0.01) | 0.80(0.03) |
| (4 MELD-Na variables) | 180 | 0.80(0.03) | 0.18(0.02) | 0.61(0.05) | 0.28(0.02) | 0.81(0.04) |
| | 90 | 0.80(0.02) | 0.16(0.01) | 0.66(0.03) | 0.25(0.01) | 0.81(0.02) |
| LR | 365 | 0.78(0.01) | 0.20(0.01) | 0.58(0.0) | 0.30(0.01) | 0.80(0.01) |
| (4 MELD-Na variables) | 180 | 0.80(0.01) | 0.18(0.01) | 0.61(0.03) | 0.28(0.01) | 0.81(0.0) |
| | 90 | 0.81(0.01) | 0.16(0.01) | 0.64(0.02) | 0.25(0.01) | 0.82(0.01) |
| RF | 365 | 0.85(0.0) | 0.22(0.02) | 0.36(0.04) | 0.27(0.02) | 0.89(0.0) |
| (4 MELD-Na variables) | 180 | 0.87(0.0) | 0.20(0.01) | 0.36(0.01) | 0.26(0.01) | 0.90(0.01) |
| | 90 | 0.89(0.0) | 0.20 (0.02) | 0.38(0.04) | 0.26(0.03) | 0.92(0.0) |

Therefore, adding additional discrete features might improve the accuracy of the MELD-Na scoring system. In addition, features of 'Reference Event-Facility' and 'Age at event' were also important features indicated by all three models, which implied the facility to which patients presented and their age at first diagnosis had strong associations with mortality.

Our study has several strengths and limitations. We were able to utilize a large database of more than 34,000 patients with cirrhosis using a retrospective design. Nevertheless, we expect our finding that these newer informatics methods better predict outcomes over the MELD-Na score in a prospective sample patient sample [32, 33]. The outcome of interest for these analyses was all-cause mortality, which we acknowledge may not always represent liver-related causes of death. Our study has another limitation. The cohort selection based on diagnosis codes (e.g., K76.6) may include patients with non-cirrhotic disease, although these conditions are frequently seen among patients with cirrhosis. Our inclusion criteria is based on the literature which was validated against chart review with good specificity [21, 34]. Furthermore, patients with primary biliary cirrhosis comprised only 216 (0.6%) of the cohort and Wilsons disease comprised an even smaller 54 (0.15%). Thus, we feel that the effect of non-cirrhotic patients in this small subset would not affect our findings. Lastly, we did have some features with large amounts of missing data requiring feature selection and imputation. This situation is commonly encountered when using clinical data for research purposes and including these cases in the pipeline improves the generalizability of the results. Our future work will further investigate patients with GI bleeding, end stage renal disease, and osteoporosis or recent bone fracture as these conditions may cause an elevated ALKP.

## Conclusions

Deep learning models (DNN) can be used to predict longer term mortality among patients with liver cirrhosis more reliably than the MELD-Na variables alone using common EHR data variables. Our findings suggest that newer informatics methods might benefit patients who are

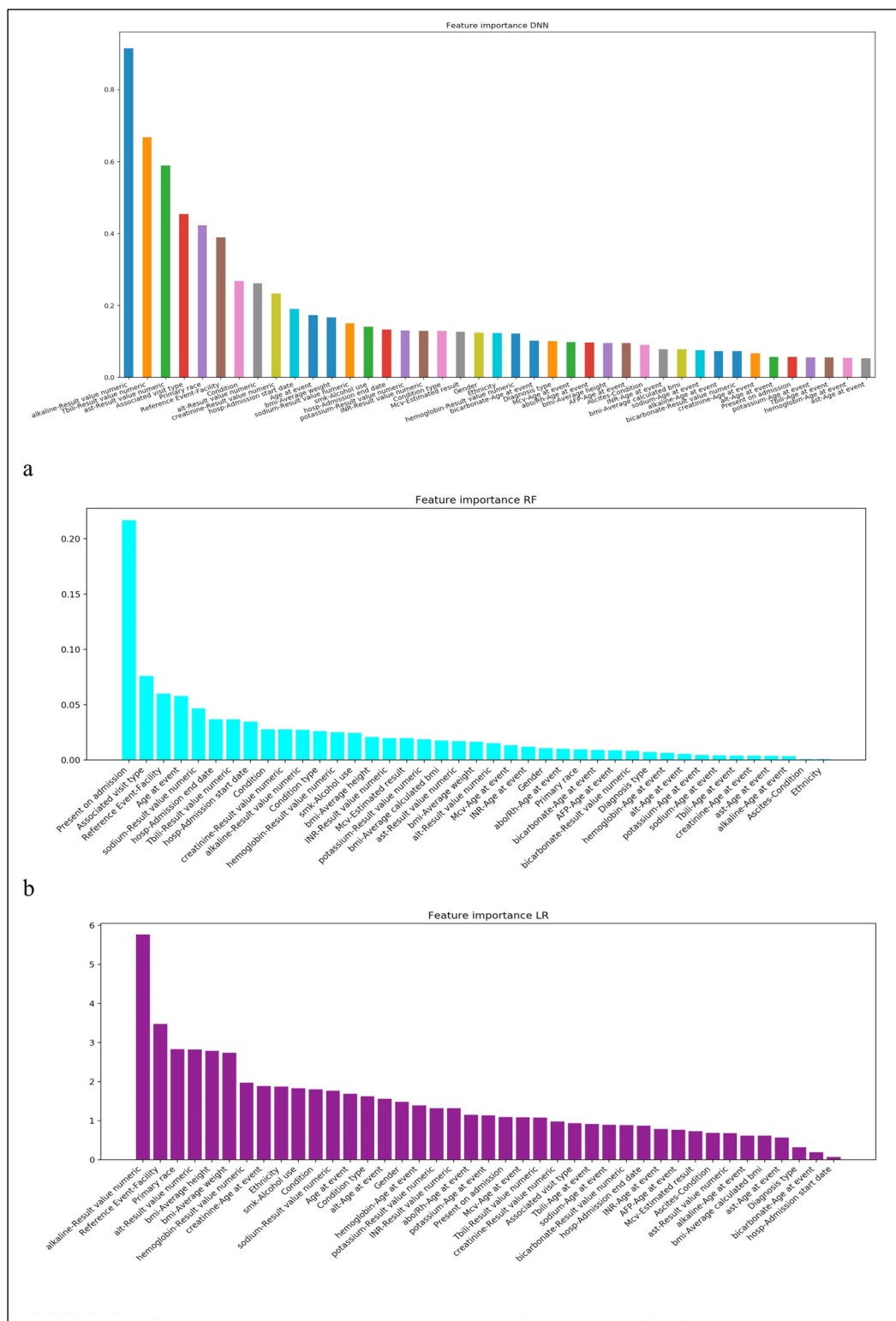

**Fig 3.** Feature importance by DNN (a), RF (b) and LR (c). This is the case for mortality within 365 days. We have similar results for 180 days and 90 days (not shown).

inadequately triaged by the MELD-Na score. Future work should validate this methodology in actual patient data and incorporate competing the competing risk of transplant to avoid mortality.

## Supporting information

**S1 Fig. Prediction performance by deep neural network (DNN), random forest (RF) and logistic regression (LR) with mean imputation strategy.** Figure a, b, c is for the case of mortality within 365 days, 180 days, and 90 days, respectively.
(PDF)

**S1 Table. The study features and feature description.**
(DOCX)

**S2 Table. Prediction metrics [n (%)] of 3 period cases for 3 machine learning models with mean strategy for imputation.**
(DOCX)

**S3 Table. Prediction metrics [n (%)] of 3 machine learning models under 10 different tradeoffs for case of 365 days.**
(DOCX)

## Author Contributions

**Conceptualization:** Randi E. Foraker.

**Formal analysis:** Aixia Guo.

**Methodology:** Nikhilesh R. Mazumder, Daniela P. Ladner, Randi E. Foraker.

**Supervision:** Randi E. Foraker.

**Writing – original draft:** Aixia Guo.

**Writing – review & editing:** Nikhilesh R. Mazumder, Daniela P. Ladner, Randi E. Foraker.

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
