## [Decision Letter · Decision Letter 0]

12 Feb 2021

PONE-D-20-37171

Predicting mortality among patients with liver cirrhosis in electronic health records with machine learning

PLOS ONE

Dear Dr. Foraker,

Thank you for submitting your manuscript to PLOS ONE. After careful consideration, we feel that it has merit but does not fully meet PLOS ONE’s publication criteria as it currently stands. Therefore, we invite you to submit a revised version of the manuscript that addresses the points raised during the review process.

We look forward to receiving your revised manuscript.

Kind regards,

Ming-Lung Yu, MD, PhD

Academic Editor

PLOS ONE

Journal Requirements:

2) Thank you for stating the following in the Acknowledgments Section of your manuscript:

[Dr. Mazumder was supported by NIH grant T32DK077662.]

 [The author(s) received no specific funding for this work.]

3)   We note that you have indicated that data from this study are available upon request. PLOS only allows data to be available upon request if there are legal or ethical restrictions on sharing data publicly. For information on unacceptable data access restrictions, please see http://journals.plos.org/plosone/s/data-availability#loc-unacceptable-data-access-restrictions.

4) Your ethics statement should only appear in the Methods section of your manuscript. If your ethics statement is written in any section besides the Methods, please move it to the Methods section and delete it from any other section. Please ensure that your ethics statement is included in your manuscript, as the ethics statement entered into the online submission form will not be published alongside your manuscript.

5) We noted in your submission details that a portion of your manuscript may have been presented or published elsewhere, as the Abstract appears in https://aasldpubs.onlinelibrary.wiley.com/doi/10.1002/hep.31579. Please clarify whether this conference proceeding/publication was peer-reviewed and formally published. If this work was previously peer-reviewed and published, in the cover letter please provide the reason that this work does not constitute dual publication and should be included in the current manuscript.

Reviewers' comments:

Reviewer's Responses to Questions

**Comments to the Author**

1. Is the manuscript technically sound, and do the data support the conclusions?

Reviewer #1: Partly

Reviewer #2: No

2. Has the statistical analysis been performed appropriately and rigorously? 

Reviewer #1: Yes

Reviewer #2: No

3. Have the authors made all data underlying the findings in their manuscript fully available?

Reviewer #1: No

Reviewer #2: No

4. Is the manuscript presented in an intelligible fashion and written in standard English?

Reviewer #1: Yes

Reviewer #2: Yes

5. Review Comments to the Author

Reviewer #1: Dear Authors, congratulations of the innovative work to improve predictive power of the prognosis of cirrhotic patients via machine learning. Here are my comments:

Q1:

Quoted from paragraph of "BACKGROUND AND SIGNFICANCE": "One 2006 study demonstrated that an artificial neural network

performed better than MELD-NA in predicting 3-month mortality among 400 patients with endstage liver disease."

 Actually MELD-NA was created by W Ray Kim in 2008, and the referred study in 2006 (Reference #10) actually compared the predicting power of liver disease‐related mortality of artificial neural network (ANN) and MELD.

Q2-0:

Quoted from paragraph of "Data source and study design": "...based on the following International Classification of Disease codes codes", the last word "codes" was repeated.

Q2-1:

The study cohort had an initial diagnosis code of liver cirrhosis in the period 1/1/2012 through 12/31/2019, however, ICD 10 codes were introduced since October 2015. What were the ICD 9 codes used to identify the study cohort?

Q2-2:

Aside from K74.3, K74.6, and K74.69, ICD 10 code K74.1 Hepatic sclerosis, and K74.2 Hepatic fibrosis with hepatic sclerosis, were also frequently used to make a diagnosis of liver cirrosis. It seemed these K74.1 and K74.2 were not included in the current study, was there a specific reason?

Q2-3:

The ICD 10 code K74.3 Primary Biliary Cirrhosis was a diagnosis used interchangeably with Primary Biliary Cholangitis. The diagnosis could be made when serum anti-mitochodrial antibody (AMA) tested positive, and evident cholangitis (elevated serum ALKP/rGT, or histologically inflammation/fibrosis of bile duct) were noted. Therefore, an ICD 10 code 74.3 may include many patients with long term cholangitis with or without treatment, but no evidently cirrhotic liver. Similar condition occurs when including E83.01 Wilson's disease, which may include those with only steatosis or chronic hepatitis. Including K74.3 and E83.01 without other ICD codes (such as K74.1, K76.6 or K74.60, etc.) may yield an inaccurate cohort for this study.

Q3:

Quoted from paragraph of Feature Extraction, "We included features that had more than 10% non-missing values, otherwise we discarded them."

10% non-missing values seemed significanly insufficient, why include features more than 10% non-missing values?

It would be more reasonable to include features with more than 90% non-missing values or less than 10% missing values.

Q4:

When performing an evaluation of cirrhotic patients, besides MELD-Na, a physician would probably choose parameters as "encephalopathy episodes" or "serum ammonia levels" (implemented in Child Pugh score, indicating liver failure if positive or elevated level), "serum albumin" (also included in Child Pugh score) or "rGT" (relates to cholestasis in cirrhosis and HCC risk in Chronic hepatitis C) and "platelet counts" (relates to liver decompensation and portal hypertension), those were parameters with established correlation to liver decompensation. In "Supplemental Table 1", Hemoglobin, Potassium and Bicarbonate were amongst the 41 features chosen for model training, what were the rationales?

Q5.

As the study revealed ALKP and Hb were among the most informative parameters for mortality prediction, did the authors excluded patients with recent GI bleeding or end stage renal disease( which may contribute to anemia) and osteoporosis or recent bone fracture( which may cause an elevated ALKP) when referencing the EHR?

Q6.

Stated in "Statistical analysis": "The LR and RF models were configured by the default options in package of Scikit-learn in Python 3.". The best hyperparameters for a Random Forest classifier were not likely to determine ahead of time, and tuning the hyperparameters to determine the optimal settings would usually be inevitable. Please specify the final configuration of LR and RF models in the current study.

Q7. Stated in "Results" of "Abstract": "The DNN model performed the best ... for 90, 180, and 365 day mortality respectively." However, in "Results" of the manuscript, it was stated that: The average AUC was 0.82 (0.79 and 0.76) for DNN model, and 0.83 (0.80 and 0.79) for RF model in the case of 90-day (180-day and 365-day) prediction for the case of 41 variables. And Figure 2 also showed RF, instead of DNN, performed the best?

Q8.

Quoted from "Results of Prediction Models": "besides these 4 variables, other features such as alkaline phosphatase values, Alanine aminotransferase values, hemoglobin values, and hospital admission start date (date difference in days between diagnosis of liver cirrhosis and previous hospital admission start dates) were also top features."

Q8-0: Why was alanine aminotransferase not mentioned in "Discussion" (Quoted: "other features such as hemoglobin, alkaline phosphatase (AP) and time since recent hospitalization were also top features and might play an important role in mortality prediction.") ?

Q8-1: According to Figure 3.(c), "hospital admission start dates" ranked least importance in LR? What were the possible explanations?

Q8-2: In Figure 3, some features, such as "Reference Event-Facility" and "Age at event", seemed to be important features in all three models, even more important than hemoglobin. Those features should be mentioned in discussion as well.

Reviewer #2: The authors seek to define an improved prognostic metric for cirrhosis using deep neural networks and machine learning. This is an important goal given limitations of the MELD/MELD-NA score.

Critiques:

- It may be helpful to cite and incorporate newer data on the MELD score. For example PMID: 31394020. This paper supports the authors’ claim that an improved prognostic metric is needed.

- The authors claim that DNN provides the best performance but it appears RF has the best AUC at each of the three time points.

- The subject selection by diagnosis would include subjects with non-cirrhotic portal hypertension. While it may be difficult to exclude such subjects, this issue should at least be addressed and mitigated if possible

- What were the causes of death in these patients? Were they liver-related? Perhaps MELD is performing poorly because it is inferior at predicting non-liver related mortality.

- The authors describe selecting features from an initial pool. How was the initial pool of features selected? Please justify why the original pool of variables may have a priori utility of prognosis in cirrhosis or discuss why they do not need any expectation of utility.

- “We included features that had more than 10% non-missing values, otherwise we discarded them” This implies that features could have up to 90% missing values. A more typical approach would be to only include features that have less than 5 or 10% missing values.

- What metric was used to assess feature importance?

- Please provide plausible/physiologic explanations for why the selected features should/could be predictive of mortality

- Please justify the use of mean/mode for missing data rather than a more sophisticated method of imputation (e.g. multiple imputation)

- Details are provided for the parameters used for DNN but not for RF and LR. "The LR and RF models were configured by the default options in package of Scikit-learn in Python 3.” It would be helpful to provide similar

- MDClone is mentioned for the first time in the discussion. This should be explained earlier.

- Table 2: How were these combinations of recall/sensitivity and specificity selected? For clinical applications it is often useful to consider set one of these metrics (sensitivity or specificity) that is expected to be clinically useful and then compare the other metric amongst the models. I recommend some consideration of the tradeoffs of sensitivity and specificity for this application depth of information for these latter models.

6. PLOS authors have the option to publish the peer review history of their article (what does this mean?). If published, this will include your full peer review and any attached files.

Reviewer #1: No

Reviewer #2: No

---

## [Author Response · Author response to Decision Letter 0]

8 Jun 2021

Predicting mortality among patients with liver cirrhosis in electronic health records with machine learning

We would like to thank the reviewers for their informed, thoughtful, and helpful comments. Please find our responses to the reviews below in italics. We believe that the manuscript has been significantly improved by our collaboration with the reviewers and hope that they will find it suitable for publication in the PlOS ONE.

Reviewers' comments:

Reviewer's Responses to Questions

Comments to the Author

1. Is the manuscript technically sound, and do the data support the conclusions?

Reviewer #1: Partly

Reviewer #2: No

Thank you for the helpful reviews. We have added more details about the technical description and conducted more rigorous experiments according to the reviewers’ comments and suggestions.

2. Has the statistical analysis been performed appropriately and rigorously? 

Reviewer #1: Yes

Reviewer #2: No

We appreciate the opportunity to improve upon our previous statistical analyses, and have conducted our experiments more rigorously according to the reviewers’ comments and suggestions. 

3. Have the authors made all data underlying the findings in their manuscript fully available?

Reviewer #1: No

Reviewer #2: No

Thank you for calling our attention to the details of this policy. Please find the data availability statement in our manuscript as follows. 

“Data availability statement

The datasets for the current study are available from the corresponding author on reasonable request. Requests to access these data sets should be directed to randi.foraker@wustl.edu.”

4. Is the manuscript presented in an intelligible fashion and written in standard English?

Reviewer #1: Yes

Reviewer #2: Yes

5. Review Comments to the Author

Reviewer #1: Dear Authors, congratulations of the innovative work to improve predictive power of the prognosis of cirrhotic patients via machine learning. Here are my comments:

Q1:

Quoted from paragraph of "BACKGROUND AND SIGNFICANCE": "One 2006 study demonstrated that an artificial neural network performed better than MELD-NA in predicting 3-month mortality among 400 patients with endstage liver disease."

 Actually MELD-NA was created by W Ray Kim in 2008, and the referred study in 2006 (Reference #10) actually compared the predicting power of liver disease‐related mortality of artificial neural network (ANN) and MELD.

We appreciate the reviewer’s insight and feedback. Our apologies for making this typo in the original manuscript. We have corrected the error.

Q2-0:

Quoted from paragraph of "Data source and study design": "...based on the following International Classification of Disease codes codes", the last word "codes" was repeated.

Thank you for pointing out this mistake. We have deleted the last word “codes”.

Q2-1:

The study cohort had an initial diagnosis code of liver cirrhosis in the period 1/1/2012 through 12/31/2019, however, ICD 10 codes were introduced since October 2015. What were the ICD 9 codes used to identify the study cohort?

This is a great question. We appreciate and agree with the reviewer’s comment. We used the MDClone platform to pull the original data as our study cohort and the platform had already implemented a crosswalk to convert ICD 9 codes to ICD 10 codes. For example, the ICD 9 code ‘572.3’ was converted to the ICD 10 code ‘K76.6’.

Q2-2:

Aside from K74.3, K74.6, and K74.69, ICD 10 code K74.1 Hepatic sclerosis, and K74.2 Hepatic fibrosis with hepatic sclerosis, were also frequently used to make a diagnosis of liver cirrosis. It seemed these K74.1 and K74.2 were not included in the current study, was there a specific reason?

We appreciate the reviewer’s insightful feedback and expert knowledge. Hepatic fibrosis and sclerosis can range from minimal to end stage, termed 'cirrhosis'. While fibrosis and sclerosis are asymptomatic, cirrhosis is not and is often associated with significant clinical complications. These late stage patients, e.g. those with cirrhosis, were the intended population for our predictive modeling, thus the other codes (K74.1 and K74.2) were not used for inclusion. 

Q2-3:

The ICD 10 code K74.3 Primary Biliary Cirrhosis was a diagnosis used interchangeably with Primary Biliary Cholangitis. The diagnosis could be made when serum anti-mitochodrial antibody (AMA) tested positive, and evident cholangitis (elevated serum ALKP/rGT, or histologically inflammation/fibrosis of bile duct) were noted. Therefore, an ICD 10 code 74.3 may include many patients with long term cholangitis with or without treatment, but no evidently cirrhotic liver. Similar condition occurs when including E83.01 Wilson's disease, which may include those with only steatosis or chronic hepatitis. Including K74.3 and E83.01 without other ICD codes (such as K74.1, K76.6 or K74.60, etc.) may yield an inaccurate cohort for this study.

Thank you for sharing all the helpful information. We appreciate the reviewer’s expert knowledge. The reviewer is correct that primary biliary cirrhosis and Wilsons disease at their early stages do not necessarily imply cirrhosis, and we have added to the discussion section to address this valid limitation to our approach. as follows. 

“Our inclusion criteria is based on the literature which was validated against chart review with good specificity. [1-2] Furthermore, patients with primary biliary cirrhosis comprised only 216 (0.6%) of the cohort and Wilsons disease comprised an even smaller 54 (0.15%). Thus, the effect of non-cirrhotic patients in this small subset would not affect our findings.”

[1] Lo Re V, Lim JK, Goetz MB, Tate J, Bathulapalli H, Klein MB, et al. Validity of diagnostic codes and liver-related laboratory abnormalities to identify hepatic decompensation events in the Veterans Aging Cohort Study. Pharmacoepidemiol Drug Saf. 2011;20: 689–99. doi:10.1002/pds.2148

[2] Chang EK, Christine YY, Clarke R, Hackbarth A, Sanders T, Esrailian E, et al. Defining a Patient Population With Cirrhosis: An Automated Algorithm With Natural Language Processing. J Clin Gastroenterol. 2016.

Q3:

Quoted from paragraph of Feature Extraction, "We included features that had more than 10% non-missing values, otherwise we discarded them."10% non-missing values seemed significanly insufficient, why include features more than 10% non-missing values? It would be more reasonable to include features with more than 90% non-missing values or less than 10% missing values.

We appreciate and agree with the reviewer’s comment. Missing values are a common issue in electronic health record data. We agree with the reviewer that 10% non-missing values seemed insufficient, but we tried to retain as many features/variables for analysis by using a low threshold (10%). We acknowledge this is a limitation and have added this limitation to the Discussion section as follows.

“Lastly, we did have some features with large amounts of missing data requiring feature selection and imputation. This situation is commonly encountered when using clinical data for research purposes and including these cases in the pipeline improves the generalizability of the results.”

Q4:

When performing an evaluation of cirrhotic patients, besides MELD-Na, a physician would probably choose parameters as "encephalopathy episodes" or "serum ammonia levels" (implemented in Child Pugh score, indicating liver failure if positive or elevated level), "serum albumin" (also included in Child Pugh score) or "rGT" (relates to cholestasis in cirrhosis and HCC risk in Chronic hepatitis C) and "platelet counts" (relates to liver decompensation and portal hypertension), those were parameters with established correlation to liver decompensation. In "Supplemental Table 1", Hemoglobin, Potassium and Bicarbonate were amongst the 41 features chosen for model training, what were the rationales?

We really appreciate the reviewer’s attention to the detail of our manuscript. Our model sought to utilize routinely obtained clinical data to improve prediction in outcomes for patients with cirrhosis. As such, we included both 'classic' liver related datapoints (creatinine, bilirubin, etc) as well as other datapoints that are routinely collected to determine if additional factors may be important (bicarbonate, hemoglobin, potassium, etc). Clinically these are routine measures and belie the severity of systemic diseases (acidosis, anemia, renin-aldosterone axis) and could provide additional accuracy to model predictions. To prevent over-fitting, we used 5-fold cross validation as described in the manuscript.

Q5.

As the study revealed ALKP and Hb were among the most informative parameters for mortality prediction, did the authors excluded patients with recent GI bleeding or end stage renal disease (which may contribute to anemia) and osteoporosis or recent bone fracture (which may cause an elevated ALKP) when referencing the EHR?

We appreciate the reviewer’s expert comments and in-depth and thoughtful feedback. The reviewer raised a great point. In this study, we did not further investigate if patients had recent GI bleeding episodes, end stage renal disease, or osteoporosis or recent bone fracture, which may cause an elevated ALKP. We have added this as a future direction in the Discussion section as follows.

“Our future work will further investigate patients with GI bleeding, end stage renal disease, and osteoporosis or recent bone fracture as these conditions may cause an elevated ALKP.”

Q6.

Stated in "Statistical analysis": "The LR and RF models were configured by the default options in package of Scikit-learn in Python 3.". The best hyperparameters for a Random Forest classifier were not likely to determine ahead of time, and tuning the hyperparameters to determine the optimal settings would usually be inevitable. Please specify the final configuration of LR and RF models in the current study.

We appreciate the reviewer’s feedback and our apologies for excluding the configuration of LR and RF models. We have added more details with regard to hyperparameter tuning as follows.

“We performed a grid search of hyperparameters for RF and LR models by five-fold cross validation. We searched the number of trees in the forest for 200, 500, and 700, and we considered the number of features for the best split according to auto, sqrt, and log2. For the LR model, we searched the norm for L1 and L2 norm in penalization, and the inverse value of regularization strength for 10 different numbers spaced evenly on a log scale of [0, 4].

 The RF model was configured as follows: the number of trees in the RF was set 500; the number of maximum features that could be used in each tree was set as the square root of the total number of features; the minimum number of samples at a leaf node of a tree was set as 1. The LR model was configured as follows: the L2 norm was used in the penalization, i.e., the variance of predicted value and real value of training data; the stopping criteria was set as 1.0*10-4; the inverse of regularization strength, which reduces the potential overfitting, was set as 1.0.”

Q7. Stated in "Results" of "Abstract": "The DNN model performed the best ... for 90, 180, and 365 day mortality respectively." However, in "Results" of the manuscript, it was stated that: The average AUC was 0.82 (0.79 and 0.76) for DNN model, and 0.83 (0.80 and 0.79) for RF model in the case of 90-day (180-day and 365-day) prediction for the case of 41 variables. And Figure 2 also showed RF, instead of DNN, performed the best?

We really appreciate the reviewer’s attention to the detail of our manuscript. Our apologies for the inconsistency of results between the Abstract and Results section. We have corrected the “Results” of the “Abstract” as follows.

“For example, the DNN model achieved an AUC of 0.88, 0.86, and 0.85 for 90, 180, and 365-day mortality respectively as compared to the MELD which only had an AUC of 0.81, 0.79, and 0.76 for the same instances.”

Q8.

Quoted from "Results of Prediction Models": "besides these 4 variables, other features such as alkaline phosphatase values, Alanine aminotransferase values, hemoglobin values, and hospital admission start date (date difference in days between diagnosis of liver cirrhosis and previous hospital admission start dates) were also top features."

Q8-0: Why was alanine aminotransferase not mentioned in "Discussion" (Quoted: "other features such as hemoglobin, alkaline phosphatase (AP) and time since recent hospitalization were also top features and might play an important role in mortality prediction.")?

Thank you. We appreciate the reviewer’s insight and feedback. Our apologies for excluding alanine aminotransferase from the “Discussion” section. We have added it to the Discussion as follows.

“Although the 4 variables used in MELD-Na model were among the top most informative features, other features such as hemoglobin, alkaline phosphatase, alanine aminotransferase, and time since recent hospitalization were also top features and might play an important role in mortality prediction.”

Q8-1: According to Figure 3.(c), "hospital admission start dates" ranked least importance in LR? What were the possible explanations?

Thank you. This is a great question aimed at possible explanations drawn by different models. One possible reason was the combination of other features already obtained higher prediction accuracy in LR.

Q8-2: In Figure 3, some features, such as "Reference Event-Facility" and "Age at event", seemed to be important features in all three models, even more important than hemoglobin. Those features should be mentioned in discussion as well.

We appreciate and agree with the reviewer’s comment. We have added text to the manuscript expanding our discussion as follows.

“In addition, features of ‘Reference Event-Facility’ and ‘Age at event’ were also important features indicated by all three models, which implied the facility to which patients presented and their age at first diagnosis had strong associations with mortality.”

Reviewer #2: 

The authors seek to define an improved prognostic metric for cirrhosis using deep neural networks and machine learning. This is an important goal given limitations of the MELD/MELD-NA score.

Critiques:

- It may be helpful to cite and incorporate newer data on the MELD score. For example PMID: 31394020. This paper supports the authors’ claim that an improved prognostic metric is needed.

We appreciate the reviewer’s insight and feedback. Our apologies for not including the newer data on the MELD score in the original manuscript. We have added the reference as follows.

Godfrey EL, Malik TH, Lai JC, Mindikoglu AL, Galván NTN, Cotton RT, et al. The decreasing predictive power of MELD in an era of changing etiology of liver disease. Am J Transplant. 2019. doi:10.1111/ajt.15559

- The authors claim that DNN provides the best performance but it appears RF has the best AUC at each of the three time points.

We appreciate and agree with the reviewer’s comment, and our apologies for the confusion. We have corrected in the Results of Abstract section with the new results as follows.

“For example, the DNN model achieved an AUC of 0.88, 0.86, and 0.85 for 90, 180, and 365-day mortality respectively as compared to the MELD which only had an AUC of 0.81, 0.79, and 0.76 for the same instances.”

- The subject selection by diagnosis would include subjects with non-cirrhotic portal hypertension. While it may be difficult to exclude such subjects, this issue should at least be addressed and mitigated if possible

Thank you. We appreciate the reviewer’s insightful feedback. We have added this limitation to the Discussion section as follows.

“Our study has another limitation. The cohort selection based on diagnosis codes (e.g., K76.6) may include patients with non-cirrhotic disease, although these conditions are frequently seen among patients with cirrhosis. Our inclusion criteria is based on the literature which was validated against chart review with good specificity.[1-2]”

[1] Lo Re V, Lim JK, Goetz MB, Tate J, Bathulapalli H, Klein MB, et al. Validity of diagnostic codes and liver-related laboratory abnormalities to identify hepatic decompensation events in the Veterans Aging Cohort Study. Pharmacoepidemiol Drug Saf. 2011;20: 689–99. doi:10.1002/pds.2148

[2] Chang EK, Christine YY, Clarke R, Hackbarth A, Sanders T, Esrailian E, et al. Defining a Patient Population With Cirrhosis: An Automated Algorithm With Natural Language Processing. J Clin Gastroenterol. 2016.

- What were the causes of death in these patients? Were they liver-related? Perhaps MELD is performing poorly because it is inferior at predicting non-liver related mortality.

We appreciate and agree with the reviewer’s insightful feedback. In our study, we stated that “The primary outcome was all-cause mortality ascertained by the medical record” with the study cohort of patients with liver cirrhosis. We have added this limitation about causes of death in the Discussion section as follows.

“The outcome of interest for these analyses was all-cause mortality, which we acknowledge may not always represent liver-related causes of death.”

- The authors describe selecting features from an initial pool. How was the initial pool of features selected? Please justify why the original pool of variables may have a priori utility of prognosis in cirrhosis or discuss why they do not need any expectation of utility.

Thank you for the reviewer’s insightful feedback. We have added the following explanations for the initial pool of feature selection.

“Baseline demographic characteristics such as age, race and ethnicity, and laboratory features collected from blood such as serum aspartate aminotransferase, alanine aminotransferase, and total bilirubin were all informative predictors for mortality predictions in patients with liver cirrhosis.[3][4]”

[3] Radisavljevic MM, Bjelakovic GB, Nagorni A V., Stojanovic MP, Radojkovicn MD, Jovic JZ, et al. Predictors of Mortality in Long-Term Follow-Up of Patients with Terminal Alcoholic Cirrhosis: Is It Time to Accept Remodeled Scores. Med Princ Pract. 2017. doi:10.1159/000451057

[4] Li Y, Chaiteerakij R, Kwon JH, Jang JW, Lee HL, Cha S, et al. A model predicting short-term mortality in patients with advanced liver cirrhosis and concomitant infection. Med (United States). 2018. doi:10.1097/MD.0000000000012758

- “We included features that had more than 10% non-missing values, otherwise we discarded them” This implies that features could have up to 90% missing values. A more typical approach would be to only include features that have less than 5 or 10% missing values.

- What metric was used to assess feature importance?

We appreciate and agree with the reviewer’s comment. Missing values are a common issue in electronic health record data. We agree with the reviewer that 10% non-missing values seemed insufficient, but we tried to retain as many features/variables for analysis by using a low threshold (10%). We acknowledge this is a limitation and have added this limitation to the Discussion section as follows.

“Lastly, we did have some features with large amounts of missing data requiring feature selection and imputation. This situation is commonly encountered when using clinical data for research purposes and including these cases in the pipeline improves the generalizability of the results.”

The metric was used to assess feature importance for each model as follows, and we have added this description to the Methods section.

“The coefficients for each input variable retrieved from the LR model was used to measure the feature importance for each input feature. The mean decrease impurity importance of a feature by the trained RF model was used to assess feature importance of RF model. We used the “iNNvestigate” package with gradient to calculate feature importance for the DNN model.”

- Please provide plausible/physiologic explanations for why the selected features should/could be predictive of mortality

Thank you for the reviewer’s insightful feedback. We have added the following explanations for why the selected features should be predictive.

“The selected features are predictive for mortality in patients with liver cirrhosis. Baseline demographic characteristics such as age, race and ethnicity, and laboratory features collected from blood such as serum aspartate aminotransferase, alanine aminotransferase, and total bilirubin were all informative predictors for mortality predictions in patients with liver cirrhosis.[3][4]”

[3] Radisavljevic MM, Bjelakovic GB, Nagorni A V., Stojanovic MP, Radojkovicn MD, Jovic JZ, et al. Predictors of Mortality in Long-Term Follow-Up of Patients with Terminal Alcoholic Cirrhosis: Is It Time to Accept Remodeled Scores. Med Princ Pract. 2017. doi:10.1159/000451057

[4] Li Y, Chaiteerakij R, Kwon JH, Jang JW, Lee HL, Cha S, et al. A model predicting short-term mortality in patients with advanced liver cirrhosis and concomitant infection. Med (United States). 2018. doi:10.1097/MD.0000000000012758

- Please justify the use of mean/mode for missing data rather than a more sophisticated method of imputation (e.g. multiple imputation)

We appreciate and agree with the reviewer’s comment. We have conducted all the analyses by using multiple imputation for all continuous variables and got better predictive values compared to using the mean strategy. So, we moved all of the results in which the mean strategy was used to the supplementary materials and replaced the main results with those using multiple imputation for missing values. The results of Figure 2 and Table 2 are attached here.

Table 2. Prediction Metrics [n (%)] of 3 period cases for 3 machine learning models.

Models Period

(days) Accuracy

Mean(std) Precision

Mean(std) Recall

Mean(std) F1-Score

Mean(std) Specificity

Mean(std)

DNN

(all variables) 365 0.83(0.01) 0.27(0.0) 0.65(0.04) 0.38(0.01) 0.85(0.01)

 180 0.86(0.02) 0.26(0.02) 0.64 (0.03) 0.37(0.02) 0.88(0.02)

 90 0.90(0.02) 0.30(0.05) 0.63(0.04) 0.40(0.04) 0.92(0.02)

LR

(all variables) 365 0.77(0.01) 0.21(0.0) 0.72(0.01) 0.33(0.01) 0.77(0.01)

 180 0.79(0.0) 0.19(0.0) 0.75(0.0) 0.31(0.0) 0.79(0.0)

 90 0.81(0.01) 0.18(0.0) 0.78(0.03) 0.29(0.01) 0.81 (0.01)

RF

(all variables) 365 0.92(0.0) 0.47(0.04) 0.37(0.02) 0.41(0.02) 0.96 (0.0)

 180 0.93(0.0) 0.46(0.03) 0.40(0.02) 0.43(0.02) 0.97 (0.0)

 90 0.94 (0.0) 0.43(0.01) 0.41(0.02) 0.42(0.01) 0.97(0.0)

DNN

(4 MELD-Na variables) 365 0.78(0.02) 0.20(0.01) 0.59(0.04) 0.30(0.01) 0.80(0.03)

 180 0.80(0.03) 0.18(0.02) 0.61(0.05) 0.28(0.02) 0.81(0.04)

 90 0.80(0.02) 0.16(0.01) 0.66(0.03) 0.25(0.01) 0.81(0.02)

LR

(4 MELD-Na variables) 365 0.78(0.01) 0.20(0.01) 0.58(0.0) 0.30(0.01) 0.80(0.01)

 180 0.80(0.01) 0.18(0.01) 0.61(0.03) 0.28(0.01) 0.81(0.0)

 90 0.81(0.01) 0.16(0.01) 0.64(0.02) 0.25(0.01) 0.82(0.01)

RF

(4 MELD-Na variables) 365 0.85(0.0) 0.22(0.02) 0.36(0.04) 0.27(0.02) 0.89(0.0)

 180 0.87(0.0) 0.20(0.01) 0.36(0.01) 0.26(0.01) 0.90(0.01)

 90 0.89(0.0) 0.20 (0.02) 0.38(0.04) 0.26(0.03) 0.92(0.0)

- Details are provided for the parameters used for DNN but not for RF and LR. "The LR and RF models were configured by the default options in package of Scikit-learn in Python 3.” It would be helpful to provide similar

We appreciate the reviewer’s feedback and our apologies for excluding the configuration of LR and RF models. We have added more details with regard to hyperparameter tuning as follows.

“We performed a grid search of hyperparameters for RF and LR models by five-fold cross validation. We searched the number of trees in the forest for 200, 500, and 700, and we considered the number of features for the best split according to auto, sqrt, and log2. For the LR model, we searched the norm for L1 and L2 norm in penalization, and the inverse value of regularization strength for 10 different numbers spaced evenly on a log scale of [0, 4].

 The RF model was configured as follows: the number of trees in the RF was set 500; the number of maximum features that could be used in each tree was set as the square root of the total number of features; the minimum number of samples at a leaf node of a tree was set as 1. The LR model was configured as follows: the L2 norm was used in the penalization, i.e., the variance of predicted value and real value of training data; the stopping criteria was set as 1.0*10-4; the inverse of regularization strength, which reduces the potential overfitting, was set as 1.0.”

- MDClone is mentioned for the first time in the discussion. This should be explained earlier.

Thank you for this suggestion. We have explained the MDClone platform for storing data earlier in the Methods section as follows.

“Our institution partnered with MDClone[5][6] (Beer Sheva, Israel) for the data storage and retrieval. MDClone platform is a data engine by storing EHR medical events in a time order for each patient. Queries can be built to pull computationally-derived or original EHR data from the platform.”

[5] Foraker R, Yu S, Michelson A, Pineda Soto J, Colvin R, Loh F, et al. Spot the Difference: Comparing Results of Analyses from Real Patient Data and Synthetic Derivatives. JAMIA OPEN.

[6] Guo A, Foraker RE, MacGregor RM, Masood FM, Cupps BP, Pasque MK. The Use of Synthetic Electronic Health Record Data and Deep Learning to Improve Timing of High-Risk Heart Failure Surgical Intervention by Predicting Proximity to Catastrophic Decompensation. Front Digit Heal. 2020. doi:10.3389/fdgth.2020.576945

- Table 2: How were these combinations of recall/sensitivity and specificity selected? For clinical applications it is often useful to consider set one of these metrics (sensitivity or specificity) that is expected to be clinically useful and then compare the other metric amongst the models. I recommend some consideration of the tradeoffs of sensitivity and specificity for this application depth of information for these latter models.

We appreciate the reviewer’s feedback and thoughtful comments. In Table 2, we used the threshold of 0.5 to calculate evaluation metrics such as recall and specificity. We agree that it is clinically useful to consider different tradeoffs of sensitivity and specificity for the specific clinical application. We have conducted the analysis considering other 10 different tradeoffs, i.e., 0.05, 0.1, 0.2, 0.3, 0.4, 0.6, 0.7, 0.8, 0.9, 0.95. The results of models with all 41 variables were summarized as Table S3 as follows for the case of 365 days.

Table S3. Prediction Metrics [n(%)] of 3 machine learning models under 10 different tradeoffs for case of 365 days.

Models Tradeoff Accuracy

Mean(std) Precision

Mean(std) Recall

Mean(std) F1-Score

Mean(std) Specificity

Mean(std)

DNN 0.05 0.52 (0.06) 0.14 (0.01) 0.93 (0.02) 0.24 (0.02) 0.48 (0.07)

 0.1 0.63 (0.06) 0.17 (0.02) 0.87 (0.04) 0.28 (0.02) 0.61 (0.07)

 0.2 0.78 (0.01) 0.22 (0.01) 0.71 (0.01) 0.34 (0.01) 0.79 (0.01)

 0.3 0.79 (0.02) 0.22 (0.01) 0.69 (0.05) 0.34 (0.02) 0.79 (0.03)

 0.4 0.82 (0.03) 0.25 (0.03) 0.62 (0.03) 0.35 (0.03) 0.83 (0.04)

 0.6 0.86 (0.01) 0.3 (0.02) 0.51 (0.03) 0.37 (0.02) 0.9 (0.01)

 0.7 0.88 (0.01) 0.31 (0.02) 0.44 (0.02) 0.37 (0.02) 0.92 (0.01)

 0.8 0.91 (0.0) 0.41 (0.03) 0.33 (0.04) 0.37 (0.03) 0.96 (0.0)

 0.9 0.93 (0.0) 0.6 (0.06) 0.21 (0.02) 0.31 (0.02) 0.99 (0.0)

 0.95 0.93 (0.0) 0.86 (0.04) 0.16 (0.03) 0.26 (0.04) 1.0 (0.0)

LR 0.05 0.25 (0.01) 0.09 (0.0) 0.96 (0.0) 0.17 (0.0) 0.19 (0.01)

 0.1 0.35 (0.01) 0.1 (0.0) 0.92 (0.0) 0.19 (0.0) 0.3 (0.01)

 0.2 0.51 (0.01) 0.12 (0.0) 0.85 (0.01) 0.22 (0.0) 0.48 (0.01)

 0.3 0.61 (0.01) 0.14 (0.0) 0.78 (0.01) 0.24 (0.0) 0.6 (0.01)

 0.4 0.7 (0.01) 0.17 (0.01) 0.71 (0.02) 0.27 (0.01) 0.7 (0.01)

 0.6 0.82 (0.01) 0.23 (0.01) 0.53 (0.02) 0.32 (0.01) 0.85 (0.01)

 0.7 0.86 (0.0) 0.27 (0.01) 0.44 (0.02) 0.33 (0.01) 0.9 (0.0)

 0.8 0.89 (0.0) 0.32 (0.01) 0.32 (0.01) 0.32 (0.01) 0.94 (0.0)

 0.9 0.91 (0.0) 0.37 (0.04) 0.15 (0.02) 0.22 (0.03) 0.98 (0.0)

 0.95 0.92 (0.0) 0.39 (0.07) 0.07 (0.01) 0.12 (0.02) 0.99 (0.0)

RF 0.05 0.51 (0.01) 0.13 (0.0) 0.93 (0.01) 0.23 (0.0) 0.48 (0.01)

 0.1 0.63 (0.01) 0.16 (0.0) 0.88 (0.01) 0.28 (0.0) 0.61 (0.01)

 0.2 0.77 (0.01) 0.22 (0.01) 0.75 (0.01) 0.34 (0.01) 0.77 (0.01)

 0.3 0.85 (0.0) 0.28 (0.01) 0.62 (0.01) 0.39 (0.01) 0.86 (0.01)

 0.4 0.89 (0.0) 0.36 (0.01) 0.5 (0.01) 0.42 (0.01) 0.92 (0.0)

 0.6 0.92 (0.0) 0.49 (0.02) 0.29 (0.02) 0.36 (0.02) 0.97 (0.0)

 0.7 0.92 (0.0) 0.53 (0.03) 0.21 (0.01) 0.3 (0.02) 0.98 (0.0)

 0.8 0.93 (0.0) 0.62 (0.03) 0.15 (0.01) 0.25 (0.01) 0.99 (0.0)

 0.9 0.93 (0.0) 0.73 (0.04) 0.11 (0.01) 0.19 (0.01) 1.0 (0.0)

 0.95 0.93 (0.0) 0.82 (0.04) 0.09 (0.01) 0.16 (0.01) 1.0 (0.0)

---

## [Decision Letter · Decision Letter 1]

22 Jul 2021

PONE-D-20-37171R1

Predicting mortality among patients with liver cirrhosis in electronic health records with machine learning

PLOS ONE

Dear Dr. Foraker,

Thank you for submitting your manuscript to PLOS ONE. After careful consideration, we feel that it has merit but does not fully meet PLOS ONE’s publication criteria as it currently stands. Therefore, we invite you to submit a revised version of the manuscript that addresses the points raised during the review process.

We look forward to receiving your revised manuscript.

Kind regards,

Ming-Lung Yu, MD, PhD

Academic Editor

PLOS ONE

Journal Requirements:

Reviewers' comments:

Reviewer's Responses to Questions

**Comments to the Author**

1. If the authors have adequately addressed your comments raised in a previous round of review and you feel that this manuscript is now acceptable for publication, you may indicate that here to bypass the “Comments to the Author” section, enter your conflict of interest statement in the “Confidential to Editor” section, and submit your "Accept" recommendation.

Reviewer #1: All comments have been addressed

Reviewer #2: (No Response)

2. Is the manuscript technically sound, and do the data support the conclusions?

Reviewer #1: Partly

Reviewer #2: Partly

3. Has the statistical analysis been performed appropriately and rigorously? 

Reviewer #1: Yes

Reviewer #2: I Don't Know

4. Have the authors made all data underlying the findings in their manuscript fully available?

Reviewer #1: No

Reviewer #2: Yes

5. Is the manuscript presented in an intelligible fashion and written in standard English?

Reviewer #1: Yes

Reviewer #2: Yes

6. Review Comments to the Author

Reviewer #1: (No Response)

Reviewer #2: The authors have largely addressed my concerns. I still question the use of features with only more than 10% non-missing values. My understanding this is contrary to common practice which would require a much higher proportion of non-missing values.

7. PLOS authors have the option to publish the peer review history of their article (what does this mean?). If published, this will include your full peer review and any attached files.

Reviewer #1: **Yes: **Ta-Wei Liu, M.D.

Reviewer #2: No

---

## [Author Response · Author response to Decision Letter 1]

25 Jul 2021

Predicting mortality among patients with liver cirrhosis in electronic health records with machine learning

We would like to thank the reviewers for their informed, thoughtful, and helpful comments. Please find our responses to the reviews below in italics. We hope that they will find it suitable for publication in the PlOS ONE.

Review Comments to the Author

Reviewer #1: (No Response)

Reviewer #2: The authors have largely addressed my concerns. I still question the use of features with only more than 10% non-missing values. My understanding this is contrary to common practice which would require a much higher proportion of non-missing values.

We appreciate and agree with the reviewer’s comment, and our apologies for the confusion in our original manuscript and the response of the first-round revision. Among all these 41 features, most of the features had low missing value rates (<20%), and only 4 features had high missing value rate (shown in the following Table 1). We tried to keep more features/variables for the study.

Table 1. Missing value rates of all studied 41 features.

Variables Non-missing value rate (%) Missing value rate (%)

Gender 100 0

Primary race 97.36 2.64

Ethnicity 67.84 32.16

Age at event 100 0

Associated visit type 86.43 13.57

Condition 100 0

Condition type 100 0

Present on admission 84.22 15.78

Diagnosis type 84.22 15.78

Ascites-Condition 100 0

hosp-Admission start date 65.42 34.58

hosp-Admission end date 65.42 34.58

bmi-Average calculated bmi 55.49 44.51

bmi-Average weight 61.45 38.55

bmi-Average height 58 42

smk-Alcohol use 48.13 51.87

Reference Event-Facility 99.99 0.01

sodium-Result value numeric 80.49 19.51

sodium-Age at event 80.49 19.51

INR-Result value numeric 62.08 37.92

INR-Age at event 62.15 37.85

creatinine-Result value numeric 79.99 20.01

creatinine-Age at event 80.06 19.94

Tbili-Result value numeric 77.01 22.99

Tbili-Age at event 77.39 22.61

Mcv-Estimated result 79.87 20.13

Mcv-Age at event 79.87 20.13

hemoglobin-Result value numeric 80.9 19.1

hemoglobin-Age at event 80.9 19.1

potassium-Result value numeric 80.56 19.44

potassium-Age at event 80.57 19.43

bicarbonate-Result value numeric 16.19 83.81

bicarbonate-Age at event 16.19 83.81

alt-Result value numeric 76.55 23.45

alt-Age at event 76.71 23.29

ast-Result value numeric 77.42 22.58

ast-Age at event 77.42 22.58

alkaline-Result value numeric 77.37 22.63

alkaline-Age at event 77.39 22.61

abo/Rh-Age at event 22.62 77.38

AFP-Age at event 14.84 85.16

---

## [Decision Letter · Decision Letter 2]

9 Aug 2021

Predicting mortality among patients with liver cirrhosis in electronic health records with machine learning

PONE-D-20-37171R2

Dear Dr. Foraker,

We’re pleased to inform you that your manuscript has been judged scientifically suitable for publication and will be formally accepted for publication once it meets all outstanding technical requirements.

Kind regards,

Ming-Lung Yu, MD, PhD

Academic Editor

PLOS ONE

Additional Editor Comments (optional):

Reviewers' comments:

Reviewer's Responses to Questions

**Comments to the Author**

1. If the authors have adequately addressed your comments raised in a previous round of review and you feel that this manuscript is now acceptable for publication, you may indicate that here to bypass the “Comments to the Author” section, enter your conflict of interest statement in the “Confidential to Editor” section, and submit your "Accept" recommendation.

Reviewer #1: All comments have been addressed

Reviewer #2: All comments have been addressed

2. Is the manuscript technically sound, and do the data support the conclusions?

Reviewer #1: Partly

Reviewer #2: Yes

3. Has the statistical analysis been performed appropriately and rigorously? 

Reviewer #1: I Don't Know

Reviewer #2: I Don't Know

4. Have the authors made all data underlying the findings in their manuscript fully available?

Reviewer #1: Yes

Reviewer #2: No

5. Is the manuscript presented in an intelligible fashion and written in standard English?

Reviewer #1: Yes

Reviewer #2: Yes

6. Review Comments to the Author

Reviewer #1: Dear authors:

There were at least five missing data rate larger than 50%, which might raise concerns:

bicarbonate-Result value numeric16.19 83.81

bicarbonate-Age at event16.19 83.81

AFP-Age at event14.84 85.16

abo/Rh-Age at event22.62 77.38

smk-Alcohol use48.13 51.87

The author could discuss more about the features with high missing values.

Firstly, what is the rationale to choose each features with large amount of missing values? Secondly, the authors may indicate the pattern of the missing data, i.e., were those data missing at completely random or not? Is there specific mechanism causing the missing data? Based on the underlying mechanism of missing data, the authors may address the model of the distribution of each feature which validates the results of multiple imputation.

Reviewer #2: (No Response)

7. PLOS authors have the option to publish the peer review history of their article (what does this mean?). If published, this will include your full peer review and any attached files.

Reviewer #1: **Yes: **Ta-Wei Liu

Reviewer #2: No

---

## [Editor Report · Acceptance letter]

23 Aug 2021

PONE-D-20-37171R2 

Predicting mortality among patients with liver cirrhosis in electronic health records with machine learning 

Dear Dr. Foraker:

I'm pleased to inform you that your manuscript has been deemed suitable for publication in PLOS ONE. Congratulations! Your manuscript is now with our production department. 

Kind regards, 

on behalf of

Dr. Ming-Lung Yu 

Academic Editor

PLOS ONE